# Biological and Cheminformatics Studies of Newly Designed Triazole Based Derivatives as Potent Inhibitors against Mushroom Tyrosinase

**DOI:** 10.3390/molecules27051731

**Published:** 2022-03-07

**Authors:** Mubashir Hassan, Balasaheb D. Vanjare, Kyou-Yeong Sim, Hussain Raza, Ki Hwan Lee, Saba Shahzadi, Andrzej Kloczkowski

**Affiliations:** 1Institute of Molecular Biology and Biotechnology, The University of Lahore, Lahore 54590, Pakistan; mubashirhassan_gcul@yahoo.com; 2The Steve and Cindy Rasmussen Institute for Genomic Medicine at Nationwide Children’s Hospital, Columbus, OH 43205, USA; 3Department of Chemistry, Kongju National University, Gongju 32588, Korea; vanjarebalasaheb@gmail.com (B.D.V.); nssj0275@gmail.com (K.-Y.S.); 4Department of Biological Science, Kongju National University, Gongju 32588, Korea; hussain_solangi@yahoo.com; 5Institute of Molecular Sciences and Bioinformatics, Nesbit Road Lahore, Lahore 54590, Pakistan; sabayseen919@gmail.com; 6Department of Pediatrics, The Ohio State University, Columbus, OH 43210, USA

**Keywords:** triazole, tyrosinase inhibitors, enzyme inhibition, kinetic mechanism, molecular docking

## Abstract

A series of nine novel 1,2,4-triazole based compounds were synthesized through a multistep reaction pathway and their structures were scrutinized by using spectral methods such as FTIR, LC-MS, 1H NMR, and 13C NMR. The synthesized derivatives were screened for inhibitory activity against the mushroom tyrosinase and we found that all the synthesized compounds demonstrated decent inhibitory activity against tyrosinase. However, among the series of compounds, N-(4-fluorophenyl)-2-(5-(2-fluorophenyl)-4-(4-fluorophenyl)-4H-1,2,4-triazol-3-ylthio) acetamide exhibited more prominent activity when accompanied with the standard drug kojic acid. Furthermore, the molecular docking studies identified the interaction profile of all synthesized derivatives at the active site of tyrosinase. Based on these results, N-(4-fluorophenyl)-2-(5-(2-fluorophenyl)-4-(4-fluorophenyl)-4H-1,2,4-triazol-3-ylthio) acetamide could be used as a novel scaffold to design some new drugs against melanogenesis.

## 1. Introduction

Melanogenesis is a process by which melanin is produced by melanocytes [1,2]. Melanocytes are present in the skin’s epidermis and hair follicles, where they are involved in melanin pigment synthesis. The skin melanocytes are surrounded by approximately 36 keratinocytes [3,4], highly specialized epithelial cells to which they transfer their synthesized melanin [5]. The melanin pigments possess anti-oxidative and photo-screening effects, therefore, they help in skin protection and wound healing [6]. However, excess production of melanin pigments can cause skin problems such as freckles, age spots, post-inflammatory hyperpigmentation, lentigo maligna, and melanoma. Ultraviolet (UV) radiation, which stimulates melanin synthesis, is reported to cause gene mutations, DNA damage, weakening of the immune system, and cancer [7]. Pigmentation in metastatic melanoma patients results in short overall and disease-free survival [7]. Melanin content is correlated with higher disease stage and seems to protect malignant melanocytes from chemo-, radio- and photodynamic therapy. Therefore, inhibition of melanogenesis could be a rational approach for controlling metastatic melanoma, abnormal skin pigmentation, and related disorders such as albinism, melasma, and vitiligo, respectively [7]. To overcome melanogenic skin related problems, medicinal chemists have been trying to synthesize new compounds by targeting the tyrosinase enzyme.

Tyrosinase is a key enzyme showing the rate-limiting effect in melanin biosynthesis. In the cytosol, L-phenylalanine may be converted to tyrosine by phenylalanine hydroxylase (PAH) to serve as the substrate for tyrosinase [8]. Tyrosinase also catalyzes the hydroxylation of L-tyrosine to 3,4-dihydroxyphenylalanine (L-DOPA) and L-DOPA to dopaquinones [9], which under unregulated conditions result in abnormal accumulation of melanin pigments [10,11]. Therefore, inhibition of tyrosinase is the simplest approach and tyrosinase inhibitors may be attractive targets to achieve depigmentation.

For melanin inhibition, potent tyrosinase inhibitors are always desirable. They can be obtained from a variety of sources, however, safety concerns pose a big challenge for their commercialization. There are several tyrosinase inhibitors, hydroquinone and kojic acid, and potent antioxidants, tert-butyl hydroxy anisole (BHA) and tert-butyl hydroxytoluene (BHT), which may show undesirable side effects, and cytotoxicity, dermatitis, and skin cancer [12,13,14,15,16]. Therefore, undoubtedly, there is an urgent need to develop more efficient and safer anti-melanogenic inhibitors using tyrosinase as target molecule. Keeping in mind the necessity and importance of inhibitory activity of triazole derivatives against tyrosinase [17], we synthesized a series of nine novel 1,2,4-triazole based compounds to explore the inhibitory potential against tyrosinase based on their functional groups attached with the parent molecule. The enzyme inhibitory kinetics of the most potent compounds was determined by Lineweaver–Burk plots and Dixon plots. Furthermore, computational molecular docking studies of the synthesized compounds against target protein PDBID 2Y9X were performed to predict the binding sites of these compounds in the target protein.

## 2. Results and Discussion

### 2.1. Synthesis

The synthesis route for the intermediates **2(a–c)**, **3(a–c)**, **5(a–c)**, **6(a–c)**, halogen-substituted/unsubstituted 2-chloro-N-phenyl acetamide derivatives **8(a–c)** and for target 1,2,4-triazole analogues **9(a–i)** are delineated in Figure 1. The compounds were synthesized by following our previously published method [17,18]. The target compound was synthesized by the coupling of 3-, 4-, and 5-substituted 1,2,4-triazole-3-thiol derivative **6(a–c)** with different 2-chloro-N-phenyl acetamide derivatives **8(a–c)** under the basic condition with a good yield. Initially, benzoic acid **1a**, 2-fluoro benzoic acid **1b**, and 4-fluoro benzoic acid **1c** were used for the synthesis of corresponding ester derivatives **2a**–**2c**. The synthesis of the ester derivatives was carried out under acidic condition. Later, aromatic ethyl benzoate functionality **2(a–c)** was changed to aromatic hydrazide **3(a–c)** by the nucleophilic substitution reaction with hydrazine hydrate in ethanol at reflux condition. Afterward, the 1,2,4-triazole scaffold was synthesized in two steps: in the first step, the subsequent hydrazide derivatives **3(a–c)** were condensed with 4-fluoro isothiocyanate to form the open ring intermediates **5(a–c)** under a nitrogen atmosphere with satisfactory yield. In the second step, the open ring intermediate derivatives **5(a–c)** were cyclized under the basic condition to form imperative key intermediates labelled as 1,2,4-triazole-3-thiol **6(a–c)**. Furthermore, another crucial intermediate was synthesized using diverse sub/unsubstituted aniline derivatives **7(a–c)** with 2-chloroacetyl chloride (2CAC) under the basic condition to form corresponding 2-chloro-N-halogen sub/unsubstituted phenyl acetamide derivatives **8(a–c)**. Finally, carbon–sulfur bond formation was carried between 1,2,4-triazole-3-thiol derivatives **6(a–c)** and 2-chloro-N-halogen sub/unsubstituted phenyl acetamide derivative **8(a–c)**. The coupling reaction was carried by using a weak base (potassium carbonate) under an inert atmosphere to form a corresponding target compound **9(a–i)** with a respectable yield. The synthesized target compounds were purified by using column chromatography techniques. The structural elucidation of the target compounds was conducted using the LC-MS, 1H NMR,13C NMR, and FTIR methods. The overall characteristic structure of the triazole derivatives is shown in Figure 1, Figure 2 and Figure 3.

#### In Vitro Tyrosinase Activity and Structural Activity Relationship (SAR)

To continue our previously published research, we designed and synthesized nine novel 1,2,4-triazole derivatives through multistep reaction pathway. The synthesized target compounds were screened for the mushroom tyrosinase inhibition study. The synthesized derivatives **9(a–i)** were modified by following our previously published work [17] to study the effect of varying substituents on a 1,2,4-triazole core on tyrosinase inhibitory activity. Our results demonstrate that the IC_50_ values for target compounds (**9a**, **9d**, **9e**, **9f**, **9g**, **9h**) ranged from 0.098 ± 0.009 to 0.379 ± 0.193 µM. However, compounds **9b**, **9c** and **9i** did not show any response against mushroom tyrosinase inhibition, as revealed in Table 1. In general, the obtained inhibitory activity in a molecule is always proportional to the contribution of the whole structure, however, the provisional SAR review was simplified by the inclusion of distinct substituents in the respective target substances. This provides more possibilities to alter the inhibitory activity of the 1,2,4-triazole analogous due to the presence of the various interaction patterns with the tyrosinase enzyme, which is reviewed and differentiated below by the SAR survey.

As displayed in Figure 1, Figure 2 and Figure 3, all target compounds **9(a–i)** were the analogues of the 1,2,4-triazole core in which the 4-position of the 1,2,4-triazole derivatives was substituted with the 4-fluoro phenyl group, which was constant throughout the series. In the case of the 5-position, it was the same only for compound **9(a–c)**, but for target compound **9(d–f)** and **9(g–i)**, the 4-fluoro phenyl group switched to the phenyl group and 2-fluoro phenyl group, respectively.

As shown in Figure 1, compounds **9a**, **9b**, and **9c** possessed identical substituents at the 4- and 5-positions, respectively. However, the para position of the N-phenyl acetamide functionality was substituted with different substituents (–H, –F or –Br etc.). In addition, all of these derivatives were screened against the mushroom tyrosinase enzyme. The acquired findings demonstrate that compound **9a** (IC_50_ = 0.124 ± 0.077 µM) exhibited good inhibitory activity, however, other derivatives (**9b, 9c**) demonstrated an ineffective counter response to mushroom tyrosinase enzyme. Therefore, the in vitro results indicate that unsubstituted N-phenyl acetamide functionality presents an exceptional outcome compared to para substitution. The target compounds **9b** and **9c** comprise a halogen group (–F and –Br) at a para position, which causes no further interaction with the enzyme, a result not detected in inhibitory activity whereas in the case of compound **9a**, which does not contain any substitution with respect to N-phenyl acetamide functionality, causing good interaction with the enzyme, and thus showing good inhibitory activity.

Additionally, in compounds **9d**, **9e**, and **9f**, the 5-position was replaced with the phenyl group, whereas the para position of the N-phenyl acetamide functionality was altered from unsubstituted **9d** (IC_50_ = 0.219 ± 0.081 µM) to halogen-substituted groups, in which compound **9e** (IC_50_ = 0.379 ± 0.193 µM) and **9f** (IC_50_ = 0.142 ± 0.068 µM) possessed fluoro and bromo substitution correspondingly. Moreover, these derivatives **9(d–f)** were subjected to screening for tyrosinase inhibition, where the findings show that all the compounds exhibited good inhibitory properties when compared with the standard drug kojic acid (IC_50_ = 16.832 ± 1.161 µM). The in vitro results suggest that when the 5-position of the 1,2,4-triazole ring was substituted with unsubstituted phenyl group **9(d–f)**, the outcomes were more satisfactory in comparison with the para fluoro-phenyl group **9(a–c)**. This suggests that all of these compounds **9(d–f)** exhibit excellent interaction with the enzyme.

Furthermore, the 4- and 5-position of the 1,2,4-triazole scaffold were substituted with 4-fluoro phenyl and 2-fluoro phenyl groups in compounds **9g**, **9h**, and **9i**, respectively, which is shown in Figure 3. Additionally, the para position of the N-phenyl acetamide functionality found in the target compounds was unsubstituted in **9g**, whereas in **9h** and **9i**, it was substituted with fluoro and bromo groups. Herewith, a compound that does not bear any substituent at the para position w.r.t N-phenyl acetamide functional exhibits good inhibitory activity (i.e., **9g** (IC50 = 0.111 ± 0.021 µM)). Additionally, when the para position of the N-phenyl acetamide functionality is replaced with the fluoro substituent such as in the case of compound **9h** (IC50 = 0.098 ± 0.009 µM), it leads to excellent inhibitory activity. Similarly, to observe the effect of inhibitory activity of compound **9i**, we replaced the fluoro groups in **9h** with a bromo group in **9i**, but the observed results were totally different. Therefore, from the in vitro results, it could be observed that the activity trend was satisfactory up to the fluoro group.

Finally, from the structure activity relationship (SAR) valuation, it was observed that many compounds displayed very good tyrosinase inhibitory activity compared to the standard drug kojic acid, except for **9b**, **9c**, and **9i**, respectively. From the results, when the 5-position of the triazole core were substituted with only the phenyl cluster and the N-phenyl acetamide functionality does not contain any substituent at the para position, such compounds showed favorable interaction pattern with the enzyme. However, when the para position of the N-phenyl acetamide functional group was exchanged with several halogens such as fluoro and bromo, then the activity trend was acceptable only up to the fluoro group, but afterward, it displayed irregularity in the inhibitory activity. The in vitro results of the synthesized derivatives were organized as per their effective interaction pattern, for instance, **9h** > **9g** > **9a** > **9f** > **9e** > **9d** (IC_50_ values are shown in Table 1 and Figure 2, Figure 3 and Figure 4).

### 2.2. Kinetic Analysis

To understand the inhibitory mechanism of synthetic compounds on tyrosinase, an inhibition kinetic study was performed. Based on our IC_50_ results, we selected our most potent compound, **9h**, to determine its inhibition type and inhibition constant. The kinetic results of the enzyme expressed by the Lineweaver–Burk plot of 1/V versus 1/[S] for different inhibitor concentrations provided a series of straight lines. The result of the Lineweaver–Burk plot for compound **9h** showed that Vmax remained the same without significantly affecting the slopes. Additionally, the Km value increased with increasing the concentration of the inhibitor. This demonstrates that compound **9h** competitively inhibits the enzyme tyrosinase, as shown in Figure 4A, and in the second plot in Figure 4B of the slope against the concentration of **9h**. The inhibition constant Ki was computed from Figure 4. B was found to be 0.018 µM.

### 2.3. Computational Analysis

#### 2.3.1. Mushroom Tyrosinase Structural Assessment

Mushroom tyrosinase, a copper-containing enzyme comprises 391 residues. The detailed structural analysis of the target protein showed that it consists of 39% of α-helices, 14% of β-sheets, and 46% of coil. The X-ray diffraction study confirmed its resolution as 2.78 Å, R-value 0.238, and unit cell crystal dimensions such as coordinate length and angles. The Ramachandran plots and values indicated that 95.90% of protein residues were present in the favored region and 100.0% residues were located in the allowed region (Appendix A, Appendix A). The Ramachandran graph values showed the good accuracy of phi (φ) and psi (ψ) angles among the coordinates of the receptor and most of the residues were plunged in the acceptable region.

#### 2.3.2. Chemo-Informatic Properties and Lipinski’s Rule of Five (RO5) Based Evaluation of Ligands

The designed ligands were analyzed computationally to predict their biological properties and RO5 validity (Table 2). The predicted chemo-informatic properties such as molecular weight (g/mol) HBD, HBA, LogP, polar surface area (PSA), molar volume and a drug-likeness score of ligand molecules were obtained. It has been confirmed from previous research data that the standard value for molecular weight is (≤500 g/mol) [19]. The computational results showed that our compounds possessed good molecular weight compared to the standard value (500 g/mol). Research data revealed that poor permeation is more likely to be observed when the HBA and HBD values exceed 10 and 5, respectively [20]. The chemo-informatics analysis showed that all of the designed compounds possessed HBA < 10 and HBD < 5. Moreover, the logP value of all compounds was also comparable with the standard value (5). All of the synthetic compounds fully obeyed the RO5. However, there are plenty of examples available for RO5 violation amongst the existing drugs [21,22].

Furthermore, the polar surface area (PSA) or total polar surface area is also known as an important parameter for drug development. The PSA parameter is commonly used for the optimization of the drug’s ability to permeate cells. Prior research data showed the standard value of PSA (<89 A2) [23]. Our analysis showed that our compounds possessed comparable PSA values. The blood–brain barrier (BBB) is also a significant parameter to interpret the activity of newly synthesized compounds. The prior reports showed that BBB values ranged (from high to low values) in the 6.0–0.0 interval, respectively. The generated results showed that all compounds possessed BBB values comparable to the standard ones. Drug-likeness is an amalgam of a complex balance of various molecular properties such as hydrophobicity, electronic distribution, hydrogen bonding characteristics, molecule size and flexibility, and the presence of various pharmacophoric features. The computational predicted results showed that most of the synthesized ligands possessed positive drug likeness values whereas ligands **9c**, **9f**, **9i**, **9j**, and **9l** exhibited negative drug likeness values. The positive and negative values prediction results showed the lead like behavior of chemical compounds and vice versa (Table 2).

#### 2.3.3. Molecular Docking Analyses

Molecular docking is a good approach to explore the binding conformation of ligands within the active site against target proteins [16,24,25,26]. The docked complexes of synthesized compounds (**9a–i**) with tyrosinase were analyzed based on the lowest binding energy values (kcal/mol) and the hydrogen/hydrophobic interaction pattern. Results showed that all the ligands (**9a–i**) exhibited comparable docking energy values compared to standard kojic acid −5.4 kcal/mol [27] and showed interaction with active site residues against mushroom tyrosinase (Figure 5). The docking energy values of all the docking complexes were calculated by using Equation (1).
∆G binding = ∆Ggauss + ∆Grepulsion + ∆Ghbond + ∆Ghydrophobic + ∆Gtors(1)
where ∆Ggauss is the attractive term for dispersion of two Gaussian functions; ∆Grepulsion is the square of the distance if closer than a threshold value; ∆Ghbond is a ramp function also used for interactions with metal ions; ∆Ghydrophobic is a ramp function; ∆Gtors is the torsional term proportional to the number of rotatable bonds. The standard error for AutoDock is reported as 2.5 kcal/mol (http://autodock.scripps.edu/, accessed on 10 February 2022). The designed molecules possessed a unique chemical skeleton with different functional group. Therefore, the synthesized ligands exhibited similar docking energy values and there was not a large energy fluctuation difference (>−2.5 Kcal/mol) among all compounds. In comparison with standard kojic values, all compounds showed good docking energy values.

##### Tyrosinase Binding Pocket Analysis and Ligand Binding Conformations

The binding pocket analysis showed that all ligands were confined in the active region of the target protein nearby Cu^2+^ metal ion. The docked complexes were superimposed to check the binding configuration of all ligands in the active region of the target protein. Results showed that the synthesized ligands bound in the binding pocket had a similar conformational pattern except for **9e** and **9l**. The **9e** and **9l** compounds bound in the upper and right side of the binding pocket, however, the most potent was bound within the active site of the target protein. All ligands showed little deviation around their axis in configuration shape and binds inside the target protein whereas the presence of different incorporated functional moieties showed their attachment inside the binding pocket near the copper metal. Most of the ligands were bound at the same position, which justified the reliability of our docking results (Figure 6A,B). In the most active ligand (**9h**), a couple of fluoro-benzene rings attached with a triazole ring showed their presence at the opening region of the binding pocket whereas other fluoro-benzene rings showed their penetration inside the binding pocket with twisted symmetry. Therefore, the result of the incorporated moiety may result in suitable configurations and conformations to ligands to be fitted inside the binding pocket of mushroom tyrosinase.

##### Hydrogen and Hydrophobic Binding Interaction

The binding interaction showed that **9h** directly binds with active region residues of mushroom tyrosinase. Binding analysis showed that **9h** forms a single hydrogen bond with His263, which provides stability to the docking complex. The fluorine atom of the fluoro-benzene ring forms a hydrogen bond (halogen) with copper bonded residue His263 with a binding distance of 3.17 Å. The aromatic His263 is a metal bonded residue that may have a significant role in the activation and functionality of tyrosinase. Our incorporated functional moiety directly indulges functional residues of the target protein, which strongly correlates with the in vitro results. The literature study suggests that binding pocket residues are significant in downstream signaling pathways [14,16,28]. The graphical depiction of the **9h** docking complex is shown in Figure 7A,B, and all other complexes in Appendix A.

## 3. Materials and Methods

### 3.1. Chemistry

All oof the chemicals required for the synthesis of intermediate as well as target compounds were procured from Sigma-Aldrich (Munich, Germany) and Samchun Chemicals (Daejeon, South Korea) and used without purification. 13C NMR and 1H NMR spectra were taken in DMSO-d6 using a Bruker Avance Ⅱ (Germany) NMR spectrophotometer at 126 and 500 MHz, respectively. The mass analysis (LC-MS) was recorded on a 2795/ZQ2000 (waters) spectrometer. The IR spectra were recorded on a Frontier FTIR spectrophotometer (PerkinElmer, Greenville, SC, USA). The progress of the chemical reactions was scrutinized by a thin layer chromatography (TLC) system. The physical factors (such as melting points) of the nine compounds denoted as **9(a–i)** were determined by using Fisher Scientific (Waltham, MA 02451, USA) melting point apparatus and were uncorrected. The coupling constant and chemical shift values were measured in ppm and Hz, respectively. Electrospray ionization (ESI) was used to produce ions for FTIR, LC-MS, 1H NMR, and 13C NMR spectral studies.

#### 3.1.1. General Synthetic Procedure for the Key Compounds **2(a–c)**, **3(a–c)**, **5(a–c)**, **6(a–c)**, **8(a–c)**, and **9(a–i)**

The synthesis of the intermediate compounds **2(a–c)**, **3(a–c)**, **5(a–c)**, **6(a–c)**, **8(a–c)**, and **9(a–i)** was performed by using our previously published method [29].

#### 3.1.2. 2-(4,5-Bis(4-fluorophenyl)-4H-1,2,4-triazol-3-ylthio)-N-phenylacetamide **(9a)**

White solid; isolated yield: 89.2%; M.P: 185.6 °C; (ESI, Appendix A): 1H NMR data: (500 MHz, DMSO-d6) δ 10.35 (s, 1H), 7.55 (ddd, J = 10.1, 9.3, 5.5 Hz, 4H), 7.46–7.37 (m, 4H), 7.37–7.27 (m, 2H), 7.28–7.16 (m, 2H), 7.11–7.01 (m, 1H), 4.19 (s, 2H); (ESI, Appendix A): 13C NMR (126 MHz, DMSO-d6) δ 165.3, 130.3, 130.2, 130.1, 130.0, 128.7, 123.4, 118.9, 117.0, 116.8, 115.7, 115.6, 36.86; (ESI, Appendix A): IR (KBr): 3064, 1734, 1669, 1601, 1228, 1158, 843, 755 cm^−1^; (ESI, Appendix A): LC-MS (*m*/*z*): calculated (422.45), found (423.2, M + 1).

#### 3.1.3. 2-(4,5-Bis(4-fluorophenyl)-4H-1,2,4-triazol-3-ylthio)-N-(4-fluorophenyl) Acetamide **(9b)**

Light grey solid; isolated yield: 86.8%; M.P: 195.3 °C; (ESI, Appendix A):1H NMR (500 MHz, DMSO-d6) δ 10.42 (s, 1H), 7.65–7.57 (m, 2H), 7.56–7.49 (m, 2H), 7.46–7.37 (m, 4H), 7.28–7.20 (m, 2H), 7.20–7.11 (m, 2H), 4.18 (s, 2H); (ESI, Appendix A): 13C NMR (126 MHz, DMSO-d6) δ 165.2, 153.6, 151.5, 135.0, 130.3, 130.3, 130.1, 130.0, 122.9, 120.8, 120.7, 117.0, 116.8, 115.7, 115.6, 115.3, 115.2, 36.7; (ESI, Appendix A): IR (KBr): 3632, 3250, 3062, 1744, 1667, 1608, 1509, 1447, 1231, 1158, 835, 822 cm^−1^; (ESI, Appendix A): LC-MS (*m*/*z*): calculated (440.44) found (441.2, M + 1).

#### 3.1.4. 2-(4,5-Bis(4-fluorophenyl)-4H-1,2,4-triazol-3-ylthio)-N-(4-bromophenyl) Acetamide **(9c)**

Dark grey solid; isolated yield: 83.1%; M.P: 219.3 °C; (ESI, Appendix A): 1H NMR (500 MHz, DMSO-d6) δ 10.50 (s, 1H), 7.58–7.47 (m, 6H), 7.41 (ddd, J = 12.2, 5.6, 2.7 Hz, 4H), 7.27–7.20 (m, 2H), 4.19 (s, 2H); (ESI, Appendix A): 13C NMR (126 MHz, DMSO-d6) δ 165.7, 153.6, 151.5, 138.0, 131.5, 130.3, 130.3, 130.1, 130.0, 122.9, 120.9, 117.0, 116.8, 115.7, 115.6, 115.0, 36.7; (ESI, Appendix A): IR (KBr): 3246, 3144, 3061, 1715.33, 1667, 1607, 1510, 447, 1228, 1098, 834 and 735 cm^−1^; (ESI, Appendix A): LC-MS (*m*/*z*): calculated (501.35), found (501.2 & 503.2, M + 1 and M + 3).

#### 3.1.5. 2-(4-(4-Fluorophenyl)-5-phenyl-4H-1,2,4-triazol-3-ylthio)-N-phenylacetamide **(9d)**

Brown solid; isolated yield: 94.4%; M.P: 206.0 °C; (ESI, Appendix A): 1H NMR (500 MHz, DMSO-d6) δ 10.36 (s, 1H), 7.57 (d, J = 8.0 Hz, 2H), 7.53 (dd, J = 7.8, 5.9 Hz, 2H), 7.39 (ddd, J = 14.4, 9.2, 4.0 Hz, 7H), 7.32 (t, J = 7.9 Hz, 2H), 7.07 (t, J = 7.3 Hz, 1H), 4.20 (s, 2H); (ESI, Appendix A): 13C NMR (126 MHz, DMSO-d6) δ 165.3, 154.3, 151.5, 138.6, 130.1, 130.0, 129.7, 128.7, 128.5, 127.8, 126.3, 123.4, 118.9, 116.9, 116.7, 36.8; (ESI, Appendix A): IR (KBr): 3076, 1670, 1554, 1510, 1225, 840, 751 and 693 cm^−1^; (ESI, Appendix A): LC-MS (*m*/*z*): calculated (404.46), found (405.3, M + 1).

#### 3.1.6. N-(4-fluorophenyl)-2-(4-(4-fluorophenyl)-5-phenyl-4H-1,2,4-triazol-3-ylthio) Acetamide **(9e)**

White solid; isolated yield: 92.1%; M.P: 220.1 °C; (ESI, Appendix A): 1H NMR (500 MHz, DMSO-d6) δ 10.42 (s, 1H), 7.64–7.56 (m, 2H), 7.56–7.49 (m, 2H), 7.45–7.38 (m, 3H), 7.37 (d, J = 4.3 Hz, 4H), 7.21–7.11 (m, 2H), 4.18 (s, 2H); (ESI, Appendix A): 13C NMR (126 MHz, DMSO-d6) δ 165.8, 154.9, 152.1, 130.6, 130.6, 130.3, 129.1, 128.4, 121.3, 121.3, 117.5, 117.3, 115.9, 115.8, 37.3; (ESI, Appendix A): IR (KBr): 3201, 3092, 1666, 1508, 1456, 1212, 834, 774 and 697cm^−1^; (ESI, Appendix A): LC-MS (*m*/*z*): calculated (422.45), found (423.2, M + 1).

#### 3.1.7. N-(4-bromophenyl)-2-(4-(4-fluorophenyl)-5-phenyl-4H-1,2,4-triazol-3-ylthio) Acetamide **(9f)**

Off-white solid; isolated yield: 90%; M.P: 247.9 °C; (ESI, Appendix A): 1H NMR (500 MHz, DMSO-d6) δ 10.50 (s, 1H), 7.58–7.48 (m, 6H), 7.40 (ddd, J = 8.7, 6.7, 2.2 Hz, 3H), 7.36 (d, J = 4.3 Hz, 4H), 4.19 (s, 2H); (ESI, Appendix A): 13C NMR (126 MHz, DMSO-d6) δ 166.1, 154.9, 152.0, 143.8, 138.6, 132.1, 130.6, 130.6, 130.5, 130.5, 130.4, 130.3, 129.1, 128.4, 121.5, 117.5, 117.3, 37.4; (ESI, Appendix A): IR (KBr): 3177, 3034, 1899, 1669, 1601, 1509, 1235, 1073, 842, 774 and 695 cm^−1^; Figure (ESI, Appendix A): LC-MS (*m*/*z*): calculated (483.36), found (485.2, M + 2).

#### 3.1.8. 2-(5-(2-Fluorophenyl)-4-(4-fluorophenyl)-4H-1,2,4-triazol-3-ylthio)-N-phenylacetamide **(9g)**

Grey solid; isolated yield: 85%; M.P: 195.1 °C; (ESI, Appendix A): 1H NMR (500 MHz, DMSO-d6) δ 10.37 (s, 1H), 7.63–7.48 (m, 4H), 7.47–7.39 (m, 2H), 7.38–7.30 (m, 4H), 7.28 (t, J = 7.6 Hz, 1H), 7.25–7.18 (m, 1H), 7.07 (t, J = 7.4 Hz, 1H), 4.24 (s, 2H); (ESI, Appendix A): 13C NMR (126 MHz, DMSO-d6) δ 165.8, 161.7, 152.0, 151.1, 139.2, 133.4, 132.5, 129.8, 129.8, 129.3, 125.3, 124.0, 119.5, 117.2, 117.0, 116.4, 116.3, 37.4; (ESI, Appendix A): IR (KBr): 3022, 1682, 1601, 1510, 1395, 1192, 1118, 903, 755 and 690 cm^−1^; (ESI, Appendix A): LC-MS (*m*/*z*): calculated (422.45), found (423.2, M + 1).

#### 3.1.9. N-(4-fluorophenyl)-2-(5-(2-fluorophenyl)-4-(4-fluorophenyl)-4H-1,2,4-triazol-3-ylthio) Acetamide **(9h)**

Dark brown solid; isolated yield: 84.2%; M.P: 217.6 °C; (ESI, Appendix A): 1H NMR (500 MHz, DMSO-d6) δ 10.43 (s, 1H), 7.59 (dd, J = 9.0, 5.0 Hz, 2H), 7.52 (ddd, J = 15.1, 14.1, 7.4 Hz, 2H), 7.43 (dd, J = 8.8, 4.8 Hz, 2H), 7.33 (dd, J = 14.6, 5.9 Hz, 2H), 7.28 (t, J = 7.6 Hz, 1H), 7.25–7.19 (m, 1H), 7.17 (t, J = 8.9 Hz, 2H), 4.22 (s, 2H); (ESI, Appendix A): 13C NMR (126 MHz, DMSO-d6) δ 165.8, 152.0, 151.1, 135.6, 133.4, 133.3, 132.5, 129.8, 125.2, 121.4, 121.3, 117.2, 117.0, 116.4, 116.3, 115.9, 115.8, 37.3; (ESI, Appendix A): IR (KBr): 3262, 3214, 3070, 1683, 1624, 1562, 1506, 1335, 1210, 1133, 966, 841 and 689 cm^−1^; (ESI, Appendix A): LC-MS (*m*/*z*): calculated 440.44, found 441.2 (M + 1).

#### 3.1.10. N-(4-bromophenyl)-2-(5-(2-fluorophenyl)-4-(4-fluorophenyl)-4H-1,2,4-triazol-3-ylthio) Acetamide **(9i)**

White solid; isolated yield: 81%; M.P: 248.9 °C; (ESI, Appendix A): 1H NMR (500 MHz, DMSO-d6) δ 10.51 (s, 1H), 7.59–7.49 (m, 6H), 7.43 (dd, J = 8.8, 4.8 Hz, 2H), 7.33 (t, J = 8.7 Hz, 2H), 7.28 (t, J = 7.6 Hz, 1H), 7.21 (dd, J = 22.7, 13.6 Hz, 1H), 4.23 (s, 2H); (ESI, Appendix A): 13C NMR (126 MHz, DMSO-d6) δ 166.1, 158.5, 151.9, 151.1, 133.4, 132.5, 132.2, 132.1, 129.8, 129.7, 125.3, 121.5, 117.2, 117.0, 116.4, 116.3, 115.6, 37.4; (ESI, Appendix A): IR (KBr): 3240, 3179, 3047, 1678, 1624, 1512, 1395, 1331, 1118, 903, 757 and 682 cm^−1^; (ESI, Appendix A): LC-MS (*m*/*z*): calculated (501.35), found (501.2 and 503.2, M and M + 2).

### 3.2. In Vitro Methodology

#### 3.2.1. Tyrosinase Assay

The inhibition of mushroom tyrosinase was determined by an alteration of the dopachrome technique using L-DOPA as a substrate [30]. The mushroom tyrosinase (EC-232-653-4) was purchased from Sigma Korea. In a detail, 140 µL of phosphate buffer (20 mM, pH 6.8), 20 µL of mushroom tyrosinase (30 U/mL), and 20 µL of the inhibitor solution were placed in the wells of a 96-well microplate. After pre-incubation for 10 min at room temperature, 20 µL of L-DOPA (3,4-dihydroxyphenylalanine, Sigma Chemical, USA) (0.85 mM) was added and the assay plate was further incubated at 25 °C for 20 min. After the incubation time, the absorbance was measured at 475 nm using a microplate reader (SpectraMax ABS, Molecular Devices, California, CA, USA) and the inhibition percentage was calculated in relation to the control. Phosphate buffer and kojic acid were tested under the same conditions as negative and positive controls, correspondingly. The amount of inhibition by the test compounds was expressed as the percentage of concentration necessary to achieve 50% inhibition (IC_50_). Each concentration was scrutinized in three independent experiments. IC_50_ values were calculated by non-linear regression using GraphPad Prism 5.0.

The % inhibition of tyrosinase was calculated as follows:Inhibition (%) = [(B−S)/B] ×100
where the B and S are the absorbances of the blank control and normal samples, respectively.

#### 3.2.2. Kinetic Analysis

Based on IC_50_, we elected the most potent compound **9h** (N-(4-fluorophenyl)-2-(5-(2-fluorophenyl)-4-(4-fluorophenyl)-4H-1,2,4-triazol-3-ylthio) acetamide) for the kinetic studies. A series of experiments were performed to regulate the inhibition kinetics of **9h** by following the already reported methods [14,31]. The inhibitor concentrations for **9h** were 0.00, 0.049, 0.098, and 0.196 µM. Substrate (L-DOPA) concentrations were between 0.0625 to 2 mM in all kinetic studies. Pre-incubation and measurement time were the same as discussed in the mushroom tyrosinase inhibition assay protocol. Maximal initial velocity was obtained from the initial linear portion of absorbance up to five minutes after the addition of enzyme at the 30 s interval. The inhibition type of the enzyme was assayed by Lineweaver–Burk plots of the inverse of velocities (1/V) versus the inverse of substrate concentration 1/[S] mM^−1^. The enzyme inhibition dissociation constant Ki was determined by the secondary plot of 1/V versus the inhibitor concentrations.

### 3.3. Computational Methodology

#### 3.3.1. Preparation of Target Protein

The crystal structure of mushroom tyrosinase with PDB ID code 2Y9X was accessed from the Protein Data Bank (PDB) (http://www.rcsb.org, accessed on 10 February 2022). The retrieved protein structure was minimized by using the conjugate gradient algorithm and AMBER force field with UCSF Chimera 1.10.1 [32]. The stereochemical properties, Ramachandran plot, and values of torsional angles [33] of mushroom tyrosinase were assessed by Molprobity server [34], while the Ramachandran graph was generated by Discovery Studio 2.1 Client [35]. The protein architecture and statistical percentage values of helices, beta-sheets, coils, and turns were assessed by the VADAR 1.8 protein structure validation web server [36].

#### 3.3.2. In Silico Design of Synthesized Compounds

The synthesized ligands **9(a–i)** were sketched by using the ACD/ChemSketch drawing package and further minimized by visualizing software UCSF Chimera 1.10.1. Molsoft (http://www.molsoft.com/, accessed on 10 February 2022) was used to predict the drug-likeness and biological properties of the designed candidate molecules. The number of hydrogen bond acceptors (HBA) and hydrogen bond donors (HBD) was also confirmed by PubChem (https://pubchem.ncbi.nlm.nih.gov/, accessed on 10 February 2022). Moreover, all the chemical compounds were validated by Lipinski’s rule of five (RO5) [37].

#### 3.3.3. Molecular Docking

The synthesized ligands were sketched in the ACD/ChemSketch tool and accessed in mol format. Furthermore, UCSF Chimera 1.10.1 tool was employed to perform energy minimization for each ligand separately using the default parameters of steepest descent steps 100 with step size 0.02 (Å), conjugate gradient steps 100 with step size 0.02 (Å), and update interval was fixed at 10. Finally, Gasteiger charges were added using Dock Prep in the ligand structure to obtain good structure conformation. A molecular docking experiment was employed on all of the synthesized ligands against α-glucosidase by using the PyRx virtual screening tool with the AutoDock VINA Wizard approach [38]. The grid box center values were adjusted as for X = −2.4528, Y = 21.4728 and Z = −31.9954, respectively. We adjusted the sufficient grid box size so it was big enough in binding pocket residues to allow the ligand to move freely in the search space. The default exhaustiveness value = 8 was adjusted in both dockings to maximize the binding conformational analysis. In all docked complexes, the ligands’ conformational poses were keenly observed to obtain the best docking results. The docked complexes were evaluated on the lowest binding energy (Kcal/mol) values and structure activity relationship analyses. The graphical depictions of all the docking complexes were carried out using Discovery Studio (2.1.0).

## 4. Conclusions

In summary, we well synthesized and characterized novel 1,2,4-triazole derivatives and screened against mushroom tyrosinase through enzyme inhibition and docking studies. The inhibition results showed that synthesized compounds possessed good inhibitory profile against mushroom tyrosinase except for **9b**, **9c**, and **9i**. Intriguingly, among all compounds, **9h** (IC_50_ = 0.098 ± 0.009 µM) exhibited promising tyrosinase inhibition activity compared to the standard drug kojic acid (IC_50_ = 16.832 ± 1.161 µM). The structure activity relationship studies revealed that **9h**, bearing the fluoro-substituent, inhibited tyrosinase effectively. Additionally, molecular docking results ascertained the good binding relationship between the mushroom tyrosinase and triazole derivatives. It was also been noted that **9h** actively binds inside the active region of mushroom tyrosinase and forms a hydrogen bond with appropriate bond length. In conclusion, **9h** has good therapeutic potential against mushroom tyrosinase. Furthermore, it can also be exploited as a potential chemical scaffold to design new drugs against melanogenesis and for clinical studies in the future.

## Data Availability

Not applicable.

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
