# Peer review of "Biological and Cheminformatics Studies of Newly Designed Triazole Based Derivatives as Potent Inhibitors against Mushroom Tyrosinase"

_molecules, 2022, doi:10.3390/molecules27051731_

Round 1
Reviewer 1 Report
In the manuscript entitled “Biological and cheminformatics studies of newly designed triazole based derivatives as potent inhibitors against mushroom tyrosinase”, authors describe the synthesis of class of triazole based derivatives compound, the evaluation of their inhibition potential on tyrosinase activity performing kinetic and computational analysis. The topic would be of interest and worth to be furthered, however, manuscript presents some critical issues regarding the results presented. List of criticisms are described below:
-table 1: SEM of some inhibitors are extremely high (in particular 9e and 9g which has a SEM value higher than the Ic50 value; is it ever acceptable to assume negative IC50 values?). Only 9h has a low SEM, is this the reason for choosing the compound 9h as the most promising? In fact, based on IC50 values, also compounds 9a, 9f and 9g could be promising compounds equally to 9h. Even more so if we consider their respective SEM values. Authors should clarify this aspect.
-line 123 “as shown in Fig 2”, is Fig.1
-line 153 “as shown in Fig 4”, is Fig.3
-line 184 “whereas the value of the Vmax remains constant with an insignificant difference”. This sentence is already explained in line 182.
-line 187, the Ki value has not any standard deviation/error
Fig 4: experimental points have no standard deviation/error. What is the unit of measurement of the SLOPE values?
-line 190. The compound is 9h, not 9e.
-did the authors perform the mushroom tyrosinase structural assessment on a purified enzyme or commercial enzyme? Is the first time that this investigation has been done on tyrosinase enzyme?
-drug-likeness score values: in lines 230-232 authors declare that all these synthetic compounds showed “good drug score values”, most of them have positive predicted values, others negative predictive values. This sentence is not clear. “These good values” are good respect to what? Is there any term of reference? Is it the same to have positive or negative values since they are considered all “good”? Also in the sentence of line 242 is reported “good docking energy values”. The definition of “good” is relative if is not supported by others studies reported in literatures which demonstrated that the drug-likeness score and the binding energy values reported by authors could be considered as “good”.
-lines 249-253: There is no clear relationship between the binding energy values reported in figure 5 and the value (or the standard error?) of 2.5kcal/mol which is considered a referring value. Why is it so important that the energy value differences among all docking complexes are lower than “a standard error value”? In addition, there is no consideration by authors regarding the fact that all these compounds have approximately the same binding energy despite wide diversity of inhibitory effectiveness.
-line 408: Authors used mushroom tyrosinase (30U/ml) for the inhibitors assays. There is no information about the enzyme used. Is it a commercial enzyme or a purified enzyme?
-Do authors considered a possible inhibition exerted by the only DMSO? No information was provided by Authors.
-line 474. “the inhibition results showed that all the synthesized compounds possessed good inhibitory profile against mushroom tyrosinase”. This sentence cannot be considered correct on the basis of results presented by the Authors. “Inhibitors 9b, 9c and 9i didn’t show any response against mushroom tyrosinase inhibition” (table 1, sentence line 105-106).
Author Response
Title: Biological and cheminformatics studies of newly designed tria- 2 zole based derivatives as potent inhibitors against mushroom 3 tyrosinase
Manuscript ID: molecules-1618436
Manuscript Type: Original Article
Reviewer 1
In the manuscript entitled “Biological and cheminformatics studies of newly designed triazole based derivatives as potent inhibitors against mushroom tyrosinase”, authors describe the synthesis of class of triazole based derivatives compound, the evaluation of their inhibition potential on tyrosinase activity performing kinetic and computational analysis. The topic would be of interest and worth to be furthered, however, manuscript presents some critical issues regarding the results presented. List of criticisms are described below:
Q1. -table 1: SEM of some inhibitors are extremely high (in particular 9e and 9g which has a SEM value higher than the Ic50 value; is it ever acceptable to assume negative IC50 values?). Only 9h has a low SEM, is this the reason for choosing the compound 9h as the most promising? In fact, based on IC50 values, also compounds 9a, 9f and 9g could be promising compounds equally to 9h. Even more so if we consider their respective SEM values. Authors should clarify this aspect.
Ans: Thank you for your deep assessment, yes 9g SEM was written mistakenly we have corrected that, and we have chosen the most potent compound based on the IC50 value not the SEM, calculation of SEM is the purpose to check how the experimental error has in the replication process that’s it.
Q2. -line 123 “as shown in Fig 2”, is Fig.1
Ans: We have replaced Fig 2 with Fig 1.
Q3. -line 153 “as shown in Fig 4”, is Fig.3
Ans: We have replaced Fig 4 with Fig 3
Q4. -line 184 “whereas the value of the Vmax remains constant with an insignificant difference”. This sentence is already explained in line 182.
Ans: We have erased the repeated sentences from the manuscript
Q5. -line 187, the Ki value has not any standard deviation/error
Ans: Yes, we have not added this, we followed our previously published protocol to add this kind of graph, however the all experiments were repeated in triplicate form.
Q6. Fig 4: experimental points have no standard deviation/error. What is the unit of measurement of the SLOPE values?
Ans: Yes as above mentioned in the answer we have not added, the unit of slope value is measured as a uM.
Q7. -line 190. The compound is 9h, not 9e.
Ans: We have corrected the compound name.
Q8. -did the authors perform the mushroom tyrosinase structural assessment on a purified enzyme or commercial enzyme? Is the first time that this investigation has been done on tyrosinase enzyme?
Ans: The structural assessment was done computational to explore the architecture of target protein and to better understand the binding pocket and overall conformation of mushroom tyrosinase. The structure was taken from PDB source having accession number 2Y9X as mentioned in the manuscript MM section.
Q9. -drug-likeness score values: in lines 230-232 authors declare that all these synthetic compounds showed “good drug score values”, most of them have positive predicted values, others negative predictive values. This sentence is not clear. “These good values” are good respect to what? Is there any term of reference? Is it the same to have positive or negative values since they are considered all “good”? Also in the sentence of line 242 is reported “good docking energy values”. The definition of “good” is relative if is not supported by others studies reported in literatures which demonstrated that the drug-likeness score and the binding energy values reported by authors could be considered as “good”.
Ans. We are thankful for such as meaningful comment for our manuscript. Therefore, we have rectified both sections as per reviewer suggestions and compare our docking results with standard kojic acid (Drug Des Devel Ther. 2015;9:4259-4268).
Q10. -lines 249-253: There is no clear relationship between the binding energy values reported in figure 5 and the value (or the standard error?) of 2.5kcal/mol which is considered a referring value. Why is it so important that the energy value differences among all docking complexes are lower than “a standard error value”? In addition, there is no consideration by authors regarding the fact that all these compounds have approximately the same binding energy despite wide diversity of inhibitory effectiveness.
Ans: Thank you very much for raising this point in the manuscript. We have rectified this section and highlighted with yellow color in the manuscript.
Q11. -line 408: Authors used mushroom tyrosinase (30U/ml) for the inhibitors assays. There is no information about the enzyme used. Is it a commercial enzyme or a purified enzyme?
Ans: Thank you for your comment, we have used commercial enzyme purchased by Sigma-Korea
Q12. -Do authors considered a possible inhibition exerted by the only DMSO? No information was provided by Authors.
Ans: Thank you for this deep assessment, Yes it’s possible to show only DMSO inhibition that’s why we have used DMSO as negative control to minus the showed inhibition.
Q13. -line 474. “the inhibition results showed that all the synthesized compounds possessed good inhibitory profile against mushroom tyrosinase”. This sentence cannot be considered correct on the basis of results presented by the Authors. “Inhibitors 9b, 9c and 9i didn’t show any response against mushroom tyrosinase inhibition” (table 1, sentence line 105-106).
Ans: We have corrected the sentence in the manuscript.
Reviewer 2 Report
Title: Biological and cheminformatics studies of newly designed tria- 2 zole based derivatives as potent inhibitors against mushroom 3 tyrosinase
Manuscript ID: molecules-1618436
Manuscript Type: Original Article
I am thankful to the journal for providing me the opportunity to review the article.
Here are my few concerns to the author regarding manuscript-
- The necessity and innovation of the article should be presented to the introduction.
- It is suggested to rewrite conclusion part in more crisp way. This section should present in one 250-300 words paragraph.
- Authors should also need to proceed for dynamical study in future path.
- There are lot of punctuation and typographical errors throughout in the manuscript. It must be rechecked by native English speaker.
- Author must be provide good high resolution/quality of picture.
- Match all the cited references in the text part.
I strongly recommend the Original article for publications in the reputed journal after minor revision.
Author Response
Title: Biological and cheminformatics studies of newly designed tria- 2 zole based derivatives as potent inhibitors against mushroom tyrosinase
Manuscript ID: molecules-1618436
Manuscript Type: Original Article
Reviewer 2
I am thankful to the journal for providing me the opportunity to review the article.
Here are my few concerns to the author regarding manuscript-
Q1. The necessity and innovation of the article should be presented to the introduction.
Ans: We have added data in the introduction section
Q2. It is suggested to rewrite conclusion part in more crisp way. This section should present in one 250-300 words paragraph.
Ans: A revised conclusion part has been added in the manuscript
Q3. Authors should also need to proceed for dynamical study in future path.
Ans: Yes, we will do further analysis in collaboration in future.
Q4. There are lot of punctuation and typographical errors throughout in the manuscript. It must be rechecked by native English speaker.
Ans: The manuscript has been rechecked by the authors and omitted all possible grammatical errors.
Q5. Author must be provide good high resolution/quality of picture.
Ans: We have attached images separately in attachment. The original figures are incorporated in the manuscript.
Q6. Match all the cited references in the text part.
Ans: We have checked all the references in the text and matched with list.
I strongly recommend the Original article for publications in the reputed journal after minor revision.
Round 2
Reviewer 1 Report
In the revised manuscript 9g SEM is not modified as described by authors in the Author's notes section.
Enzyme source has to be added in the Materials and Methods section.
Author Response
Q1. In the revised manuscript 9g SEM is not modified as described by authors in the Author's notes section.
Ans: We really excused our mistake, now 9g SEM value has been rectified in the revised manuscript.
Q2. Enzyme source has to be added in the Materials and Methods section
Ans: The enzyme source with EC number has been added into the methodology section.